# Fabrication and Performance of Graphene Flexible Pressure Sensor with Micro/Nano Structure

**DOI:** 10.3390/s21217022

**Published:** 2021-10-23

**Authors:** Weibin Wu, Chongyang Han, Rongxuan Liang, Jian Xu, Bin Li, Junwei Hou, Ting Tang, Zhiheng Zeng, Jie Li

**Affiliations:** 1College of Engineering, South China Agricultural University, Guangzhou 510642, China; wuweibin@scau.edu.cn (W.W.); 20202009003@stu.scau.edu.cn (C.H.); samleung@stu.scau.edu.cn (R.L.); xujianaa@stu.scau.edu.cn (J.X.); 20203163056@stu.scau.edu.cn (T.T.); zengzhiheng@stu.scau.edu.cn (Z.Z.); lj@stu.scau.edu.cn (J.L.); 2College of Intelligent and Manufacturing Engineering, Chongqing University of Arts and Sciences, Yongchuan, Chongqing 402160, China; 20210011@cqwu.edu.cn

**Keywords:** laser-induced graphene, flexible pressure sensor, micro-nano structure, sensitivity

## Abstract

Laser-induced graphene (LIG) has been widely used in flexible sensors due to its excellent mechanical properties and high conductivity. In this paper, a flexible pressure sensor prepared by bionic micro/nanostructure design and LIG mass fraction regulation is reported. First, prepared LIG and conductive carbon paste (CCP) solutions were mixed to obtain a conductive polymer. After the taro leaf structure was etched on the surface of the aluminum alloy plate by Nd:YAG laser processing, the conductive polymer was evenly coated on the template. Pressure sensors were packaged with a stencil transfer printing combined with an Ecoflex flexible substrate. Finally, the effects of different laser flux and the proportion of LIG in the composite on the sensitivity of the sensor are discussed. The results show that when the laser flux is 71.66 J·cm^−2^ and the mass fraction of LIG is 5%, the sensor has the best response characteristics, with a response time and a recovery time of 86 ms and 101 ms, respectively, with a sensitivity of 1.2 kPa^−1^ over a pressure range of 0–6 kPa, and stability of 650 cycle tests. The LIG/CCP sensor with a bionic structure demonstrates its potential in wearable devices.

## 1. Introduction

Since Smith et al. discovered the piezoresistive effect of silicon and germanium in 1954 [1], pressure sensors have entered a stage of rapid development. Various new flexible pressure sensors have attracted wide attention in the fields of physiological condition detection and medical diagnoses [2,3,4,5]. Currently, research on improving the performance of flexible pressure sensors is mainly carried out by modifying the conductive sensing layer of the sensor through the introduction of microstructures [6], followed by the preparation of high-performance composite materials, (such as graphene) and the selection of materials to improve the sensitivity [7], stability [8], and responsiveness of flexible devices [9]. Among these materials, laser-induced graphene (LIG) has been widely studied and applied in wearable flexible sensors due to its advantages of simplicity, rapid fabrication, and low cost [10].

With the rapid development of laser direct writing technology, more and more scholars use laser-assisted technology to prepare graphene on flexible carbon substrates due to the advantages of being simple, mask-free, large-scale, and low cost. Furthermore, the graphical manufacturing process and unique properties (excellent electrical conductivity and stability) of LIG have led to an increasing number of applications in intelligent sensing [11]. Papazoglou et al. designed an alternative printing approach based on the laser induced forward transfer (LIFT) technique for the successful digital transfer of graphene, the 2D material par excellence [12]. Liu et al. proposed an upgraded process for assembling multifunctional graphene coatings on polymeric composites by combining extrusion-printing and LIG. Continuous optimization of laser-assisted technology has led to better quality graphene [13]. For LIG fabrication, polyimide (PI) can be irradiated with a carbon dioxide (10.6 μm) infrared laser instantaneously to cause local high temperatures and carbon atom rearrangement into a porous graphene structure by breaking carbon–oxygen bonds and carbon–nitrogen bonds. However, selection of appropriate precursor materials and improvement of the manufacturing strategy are key to improving the performance of the sensors [14,15,16]. By the casting method, Gong et al. developed an ultrathin sensor with a response time of less than 22 ms by using a gold film with a thickness of 1.64 μm on a PDMS elastomer substrate. However, the cost of gold film is high, which has a negative impact on the cost of devices and systems [17]. Boland et al. added different volume fractions of graphene to a highly viscoelastic polymer, the resulting nanocomposites display unusual electromechanical behavior, such as postdeformation temporal relaxation of electrical resistance and nonmonotonic changes in resistivity with strain [18]. Carvalho et al. obtained a strain sensor by transferring raw LIG from a PI precursor substrate to a PDMS elastomer, which has a strain coefficient of up to 20,000 and a tensile property of 100% and can be used to detect finger motion [19]. In terms of improving the manufacturing process, Pan et al. developed a flexible electronic skin pressure tactile sensor with pressure resistance by improving the shape of the microstructure. The sensor can monitor low pressure below 1 Pa but still has high sensitivity in the high-pressure area [20]. Tao Yong et al. proposed a graphene paper pressure sensor with excellent performance in the range of 0–20 kPa, which can be used in pulse detection, respiration detection, and other fields [21]. Therefore, exploring manufacturing strategies of different micro/nanostructures and other functional materials is advisable [22].

Despite the recent use of LIG in the flexible application of wearable devices with good results, there are many problems, such as unstable sensitivity and skin sweat; the materials and structure still need to be further improved. Taro leaves have superhydrophobic surface structure [23], under the observation of scanning electron microscopy (SEM), the hexagonal protrusions and petal-like nanostructures on a micron scale are evenly distributed in the corresponding nest-shaped caves [24]. Many tiny needles are harmoniously distributed on the surface, which has a positive effect on improving the contact response of the flexible sensor. On the other hand, high-quality LIG is usually available on thermoplastic polymers with a high melting point [25,26,27]. Compared with flexible elastomers, PI usually shows a high rigidity and poor tensile properties. Therefore, an additional transfer process is required to manufacture LIG-based devices on flexible elastic substrates [28,29]. Conductive carbon paste (CCP) is widely used in the electronic packaging industry for its excellent electrical conductivity and stability [30]. In this paper, we used SEM to extract the biological template of taro leaves and mastered the laser processing of the graphene double-scale bionic structure. Conductive polymers were prepared by mixing LIG and CCP in different mass fractions. Then, CCP/LIG films with taro leaf structures of different aspect ratios inverted from templates obtained by laser carving were fabricated. By combining the film with an Ecoflex flexible substrate, a flexible pressure sensor was assembled. Finally, we discuss the relationship between the sensitivity of the sensor and different aspect ratios of CCP/LIG and proportions of LIG in the composite material caused by changes in the laser flux; the sensor cycling stability, response time, and response were tested at different frequencies.

## 2. Materials and Methods

### 2.1. Materials

The Conductive carbon paste (JC-2110, from China) was mixed with LIG to prepare Conductive polymers due to excellent electrical conductivity and stable resistance performance. SEM (S-3700N, acceleration voltage: 0.3–30 kV, Hitachi, Japan) was used to analyze the surface morphology of LIG. In addition, a three-dimensional surface topography analyzer (BMT Expert, BMT GmbH, Rostock, Germany) was used to characterize the surface roughness and ablation degree of the samples. A Raman spectrometer (Lab RAM HR800, 50× objective, laser beam less than 10 μm) was used to characterize the atomic structure of LIG. The resistance (FFT-331, Rooko Electrical, Hyogo, Japan) of LIG/CCP was measured using a four-probe square resistance tester. The pressure test of the sensor was carried out with a binding force tester (Ag-X Plus, 100 N). A digital source meter (KEITHLEY-245, Cleveland, OH, USA) was used to monitor the resistance changes of the pressure sensors in real time.

### 2.2. Preparation of a Porous Graphene Sensor with a Bionic Structure

The preparation process of the flexible pressure sensor is shown in Figure 1 and Figure 2. The details are as follows:

First, the microstructure of the taro leaf surface was imaged by SEM in Figure 3c, and the regular structure of the taro leaf was designed for imitation. An aluminum alloy plate with a size of 30 mm × 10 mm × 3 mm was selected, cleaned by ultrasonication, and dried [31]. After that, the aluminum alloy plate was carved into the form of the taro leaf surface with different laser injection amounts by an Nd:YAG laser. PI film with a thickns of 125 μm was pasted on a glass plate with a thickness of 50 mm as the substrate. Subsequently, graphene was successfully prepared on the PI surface by an Nd:YAG laser two-step induction method. The Raman spectrum and an SEM image of the prepared LIG are shown in Figure 1b,c. The PI film we use comes from DuPont-Toray Co. Ltd. (Tokyo, Japan) and has a thickness of 125 microns. Moreover, the Ecoflex (0050, Smooth-On Inc., Macungie, PA, USA) we selected is a soft silicone rubber material with excellent elasticity, ductility, and water resistance. The laser equipment used was a computer-controlled laser engraving machine (CLS8100), which was equipped with a diode-pumped Nd:YAG solid-state pulse laser with an output central wavelength of 1064 nm, a maximum output power of 20 W, a repetition rate of 50 kHz, and a pulse duration of 150 ns. The focusing diameter of the laser beam was 40 μm and the repetition accuracy was ±2.5 μm.

The process of experiment as follows, n-hexane was used as a diluent, in which CCP was dissolved and diluted for 1 h under magnetic mixing at 750 r/min. Then, LIG was added to the diluted CCP solution in different mass fractions to prepare CCP and LIG mixed solutions with mass fractions of 0%, 3%, 5%, 7%, and 9%, as shown in Figure 2a. The dispersed CCP/LIG composite solution was placed in an ultrasonic cleaning machine for 2 h of deep dispersion treatment. Finally, a uniform CCP/LIG dispersion was obtained. CCP/LIG dispersions were poured into an aluminum alloy plate with different microstructures and defoamed in vacuum. After natural cooling and solidification, CCP/LIG films with different scales of taro leaf structure modification were obtained. The size of CCP/LIG films was 30 mm×10 mm. Both ends of the CCP/LIG films were bonded with conductive silver and soft copper-clad PI film. Moreover, the packaging of the flexible CCP/LIG sensor was completed by two sensing layers facing each other, supported by Ecoflex as the substrate, as shown in Figure 2b.

## 3. Results and Discussions

### 3.1. Characterization of Micro/Nanostructures

On the one hand, it is easier to produce remarkable structure on regions using parallel line laser paths than on regions using grid line laser paths [32]. On the other hand, different laser scanning rates, powers, and processing times not only affect the morphology of the etched microstructure, but also affect the size of the microstructure by changing the machining depth [31,33]. Therefore, a parallel laser processing path (with an interval of 0.01 mm and the laser scanning process repeated 10 times) was adopted to obtain a surface with greater depth. The laser scanning speed was 500 mm/s, and the static laser flux (i.e., the laser power) was selected as 15.92 J·cm^−2^, 31.85 J·cm^−2^, 47.77 J·cm^−2^, and 55.73 J·cm^−2^; three samples were prepared for each laser injection rate. As shown in Figure 3b, with increasing laser flux, the height of the microstructure protrusion increases. When the laser flux is 71.66 J·cm^−2^, the height of the protrusion is 0.33 mm and the average roughness is 0.0777 mm. As more laser pulses ablate the surface and remove more material, more splash structures accumulate at the edge of the unmachined area, which leads to an increase in the microstructure protrusion height. Thus, the microstructures generated by laser ablation can increase the surface roughness of the aluminum alloy.

To further characterize the micro/nanostructure of the samples, we collected a group of CCP/LIG films transferred onto bionic structure templates obtained with different laser processing parameters. Figure 3a shows the SEM image of a taro leaf surface. Figure 3c shows the CCP/LIG microstructure image transferred from an aluminum alloy template when the laser flux is 71.66 J·cm^−2^. The height of the protrusion is 0.33 mm and the average roughness is 0.0777 mm, the same as in Figure 3b. Figure 3d shows the black dotted section of CCP/LIG microstructure. It is proven that a CCP/LIG surface with intact taro leaf microstructure can be successfully prepared by laser engraving and transfer printing.

Graphene materials contain a large number of C-C covalent bonds and have a symmetric structure; the symmetric molecules and covalent bonds have fixed vibration frequencies. The samples were characterized by a dispersive confocal Raman spectrometer with a 532 nm wavelength laser [34,35] in Figure 4a1–d1. The peak at 1609 cm^−1^ (G peak) and the peak at 1348 cm^−1^ (D peak) represent the in-plane vibration mode of an SP^2^ hybrid carbon atom and the ring breathing vibration mode of a hybrid carbon atom ring in graphene, respectively. The D peak is characterized by the defect and disorder of a carbon lattice [36]. As shown in Figure 4, with the increase of LIG content, the intensity of G peak and D peak gradually decreases, indicating that the CCP/LIG intermediate layer with low LIG content is mainly multilayer graphene material [37]. In particular, the D peak of low LIG content was particularly intense, indicating more defects in the prepared samples, while the D peak of high LIG content (7%) was extremely weak, accompanied by better effect and less layers of graphene. Moreover, the G peak of low LIG content is much stronger than that of high LIG content (7%), and the G peak shape is very sharp and sensitive to the number of graphene layers. In addition, the 2D peak represents the vibrational pattern of the two photonic lattices, is also the frequency multiplier of peak D, and it is always very strong in both low and high LIG content samples. A comprehensive judgment can be made that the CCP/LIG interlayer materials with high LIG contents are low layer graphene material with better effect.

### 3.2. Performance Test of the Flexible Pressure Sensor

In this section, the LIG/CCP sensing layer of the bionic taro leaf structure is taken as the research object to explore the relationship between the sensitivity of the sensor, different laser injection amounts, and different LIG mass fractions. The sensitivity is the key factor of the sensor; its formula is expressed as (*R* − *R*_0_)/(*R*_0_*P*), where the real-time resistance is expressed as *R*, the initial resistance is expressed as *R*_0_, and the pressure on the sensor surface is expressed as *P* [38].

#### 3.2.1. Influence of the Laser Flux on the Sensor

Figure 5a clearly shows the influence of the taro-like leaf structure formed under different laser fluxes on the resistance change rate-pressure curves. The data show that the sensitivity of the sensor changes with the static laser scanning flux. Significant changes occur, mainly because the laser scanning flux directly determines the size/height of the bionic structure and the surface roughness [39].

Comparing different laser fluxes, when the laser flux is at a maximum, that is, 71.66 J·cm^−2^, the taro-like aluminum alloy plate is subjected to more intense laser irradiation, and, simultaneously, a more intense ablation phenomenon is produced in the laser operation area such that the taro-like aluminum alloy plate has a higher aspect ratio. Similarly, the CCP/LIG conductive sensing layer obtained through template transfer will also exhibit the bionic surface structure with the highest aspect ratio in the experimental group. As shown in Figure 3, when the laser injection amount is 71.66 J·cm^−2^, the corresponding CCP/LIG film has a larger longitudinal structure, and with the increase in the laser injection amount, the longitudinal size of the microconvex structure on the taro-like CCP/LIG surface increases accordingly. When the laser injection amount for Group A is 15.92 J·cm^−2^, the height of the convex structure is 140 μm. When the laser injection amount for Group B is increased to 31.85 J·cm^−2^, the height of the convex structure is 200 μm, and the size of the convex structure is increased by 42.9% compared with that of Group A, which is the largest increase in the microconvex structure scale among the adjacent groups. According to Figure 5, Group B has the largest improvement in sensitivity relative to Group A compared with other experimental groups. With the increase in the laser injection amount to 47.77 J·cm^−2^, 55.73 J·cm^−2^, and 71.66 J·cm^−2^, the ablation effect of the laser continues to increase, and the raised height of the taro leaf structure reaches 230 μm, 260 μm, and 330 μm, respectively, with a gradual improvement in sensor sensitivity. The relationship between sensor properties and machining parameters is shown in Table 1.

When the CCP/LIG sensor is subjected to a certain load, the sensor in Group E has a smaller initial contact area, and, when the two CCP/LIG layers are affected by the load, a larger shape deformation can be more easily generated in Group E than in the other groups. Furthermore, due to the smaller initial contact area of the sensor in Group E, its initial resistance, R0, is lower in the initial state, which eventually leads to an increase in the sensitivity of CCP/LIG as the laser injection amount increases. The sensitivity curve of the CCP/LIG flexible pressure sensor can be divided into three stages in the stress application process: (1) When no load is applied to the sensor, because the CCP/LIG contact surface is supported by the taro leaf structure, the contact area is small, and the resistance is large. (2) When a partial load is applied to the sensor, with an increase in load, the two CCP/LIG pieces quickly undergo occlusion. At this stage, due to the high speed of the change in the contact area, the resistance rapidly decreases, which is characterized by a high resistance change rate and a high sensitivity. (3) A greater load continues to be exerted on the sensor at this time because in the second stage, the two pieces of CCP/LIG have reached a higher occlusion level, meaning that a relatively stable conductive path is formed. As the load continues to be exerted, the conductive path change is smaller, so less resistance change is reflected on the macroscale, as shown in Figure 5a.

#### 3.2.2. Influence of the LIG Content on the Sensor

We chose the CCP/LIG obtained when the laser injection amount was 71.66 J·cm^−2^ as the research object and tested the influence of LIG on the conductivity of the conductive sensing layer and the sensor by changing the proportion of LIG in CCP/LIG.

Figure 5b shows the influence of the LIG content on the resistivity (average value) of the conductive sensing layer of the sensor. We designed five samples with different LIG contents; the error bar represents standard deviation. The results show that when the mass fraction of LIG in CCP/LIG increases from 3% to 5%, the resistivity of CCP/LIG sharply decreases. When the LIG content is 3%, the resistivity of CCP/LIG is 14.58 Ω.mm. When the LIG content is 5%, the resistivity of CCP/LIG is reduced to 7.94 Ω.mm, and the electrical conductivity is increased by 45.5%. With increasing LIG content from 5% to 7% and 9%, the CCP/LIG resistivity continues to decrease, but the rate of decline is greatly reduced. When the LIG content is 7%, the CCP/LIG resistivity is 7.28 Ω.mm, compared with 5% LIG content, where the CCP/LIG resistivity only decreases by 9%. When the LIG content is 9%, the resistivity of CCP/LIG is 7.02 Ω.mm, which is a decrease of 3.6% compared with 7%. In conclusion, within a certain range, CCP/LIG shows better conductivity with increasing LIG content, but when the LIG content exceeds a certain threshold, the conductivity of CCP/LIG gradually becomes stable. As shown in Figure 4, when the LIG mass fraction is increased to 5% and 7%, the SEM morphology greatly changes, which is embodied in the increases in the petal-like dispersed matter and the roughness of the surface. Moreover, a more complex staggered conductive network is formed. Therefore, this characterization also provides effective support for the resistance change of the conductive sensing layer of the CCP/LIG sensor.

Figure 5c depicts the CCP/LIG sensor sensitivities for different LIG mass fractions. When the mass fraction of LIG increases from 0% to 5%, the sensitivity of the CCP/LIG sensor continuously increases. When the mass fraction of LIG reaches 5%, the sensitivity of the CCP/LIG sensor is the highest. As the mass fraction of LIG continues to increase, the conductivity of CCP/LIG continues to increase, but the sensitivity of the sensor shows a downward trend. The higher the conductivity is, the lower the sensitivity. At this time, the sensitivity and conductivity show a negative correlation. When LIG is in the 0% to 5% range, we think that the cause of the above phenomenon is likely that CCP/LIG has a more conductive network with defects, and an increase in the LIG content in this range has a larger impact on the conductive network defects. LIG nanostructures are speculated to be able to repair the defects at the same time to rapidly improve the CCP/LIG network [40]. According to the experimental data, when the amount of LIG is 5 wt%, the conductive network can be repaired to a higher degree, and the conductivity of CCP/LIG reaches the bottleneck area. When the LIG content continues to increase from 5% to a higher level, significant improvement of the conductive network of CCP/LIG by LIG is difficult. At the same time, due to the small increase in conductivity, the initial resistance is low, which also results in a negative correlation between the sensitivity and conductivity. At this stage, due to the good conductivity of CCP/LIG itself, the contribution brought by the continuous improvement in the conductive network morphology cannot compensate for the impact of the initial resistance reduction, so the sensitivity decreases with increasing LIG content.

Figure 6a shows the resistance change rate diagram and response time diagram of the flexible pressure sensor when a uniform load is applied from the outside, and samples with 5% LIG content were selected for the test. As seen from the figure, when the uniform load is loaded, the response times of the CCP/LIG sensor for each deformation remain highly consistent, the response time reaches 86 ms, the recovery time is 101 ms when the load is removed, and the difference between the response time and the recovery time is 15 ms. The time lag when the external load is unloaded is assumed to be due to the slight viscosity of the Ecoflex, which causes the two CCP/LIG pieces to not be fully restored to the initial contact state. This phenomenon is common in various flexible sensors.

We examined three groups of loading tests with different frequencies: 0.10 Hz (gray); 0.16 Hz (yellow); and 0.24 Hz (blue). Under a constant pressure of 10 kPa, we carried out loading tests on the sensor to verify whether the sensor would be affected by the frequency and produce signal errors. As shown in Figure 6b, when the sensor is affected by the same load at different frequencies, its sensitivity level maintains a high consistency, and the response of the sensor at various frequencies also maintains excellent stability. We carried out repeated loading–unloading cycle tests on the sensor at a constant pressure of 10 kPa. The sensor was tested every 50 cycles as a test point, and the data shown in Figure 6c were obtained. After 650 tests, the sensor still shows stable response characteristics, with no significant changes in sensitivity and no attenuation. The flexible CCP/LIG sensor with Ecoflex as the flexible support substrate is shown to have a long working life and high stability during the working life. As the hysteresis effect is a significant characteristic in pressure sensors, it follows that further investigation of the hysteresis effect is needed.

## 4. Conclusions

We synthesized a conductive polymer by blending LIG with conductive carbon paste (CCP) and fabricated a CCP/LIG-based flexible pressure sensor through the design of a taro leaf structure and control of the mass fraction of LIG. Since the sensitivity is not only affected by the microstructure but also related to the conductivity of the sensing layer, the sensitivity of the sensor is greatly improved by adjusting the laser injection amount and the mass fraction of LIG in the polymer. When the laser flux is 71.66 J·cm^−2^ and the mass fraction of LIG is 5%, the optimal sensitivity of the sensor is 1.2 kPa^−1^. In addition, we use Ecoflex as a flexible substrate to encapsulate the sensor face-to-face due to Ecoflex’s excellent elasticity and tensile properties. The response time and recovery time of the sensor are 86 ms and 101 ms, respectively. The sensor has high reliability and sensitivity: a stability of 650 cycles under a constant pressure of 10 kPa and different frequencies (0.10 Hz, 0.16 Hz, and 0.2 Hz).

In summary, this flexible pressure sensor with bionic structure shows good performance compared with sensors prepared by chemical synthesis methods (such as CNF/SBS and PDMS/MWCNTs, etc.), showing great application potential in wearable electronic devices, medical monitoring equipment, and human–machine interface.

## Figures and Tables

**Figure 1 sensors-21-07022-f001:**
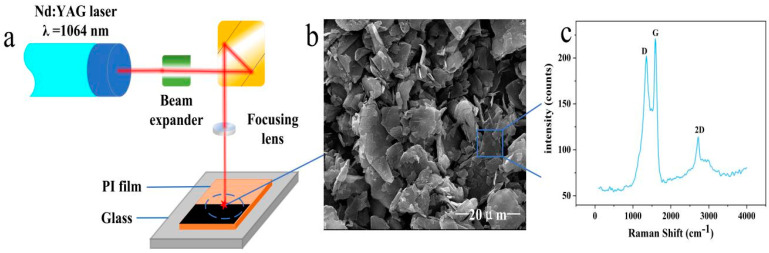
Process to prepare LIG: (**a**) schematic illustration of LIG preparation; (**b**) SEM image of graphene; (**c**) Raman spectrum of graphene.

**Figure 2 sensors-21-07022-f002:**
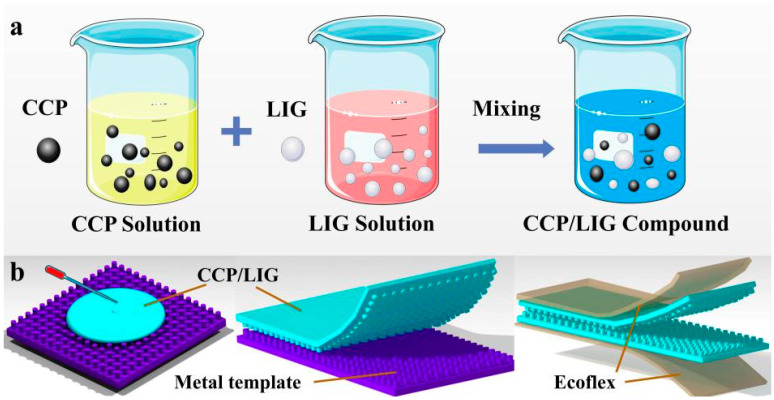
Manufacturing process of the flexible pressure sensor: (**a**) preparation of CCP/LIG dispersions; (**b**) schematic diagram of the preparation of the flexible pressure sensor.

**Figure 3 sensors-21-07022-f003:**
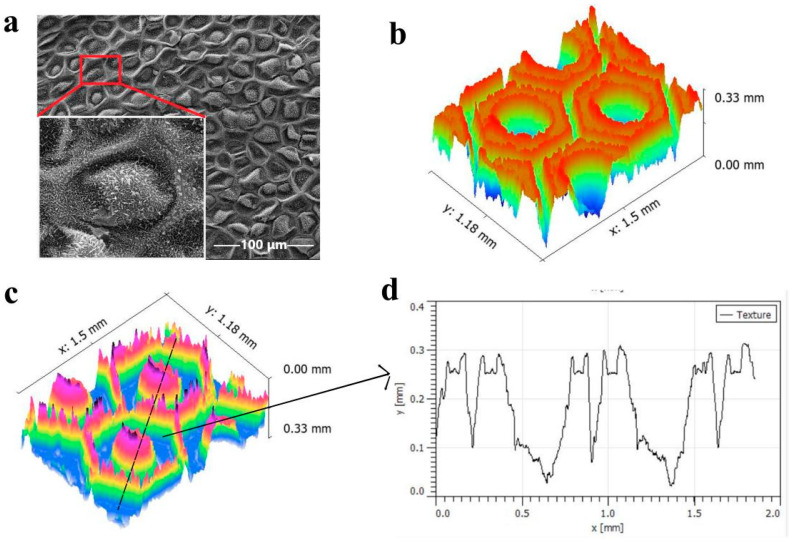
(**a**) The SEM image of a taro leaf surface; (**b**) The 3D morphology of the processed metal template when the laser flux is 71.66 J·cm^−2^; (**c**) The 3D morphology of the CCP/LIG film obtained by transfer printing of the metal template processed with a laser flux of 71.66 J·cm^−2^; (**d**) Sectional profile of the CCP/LIG film.

**Figure 4 sensors-21-07022-f004:**
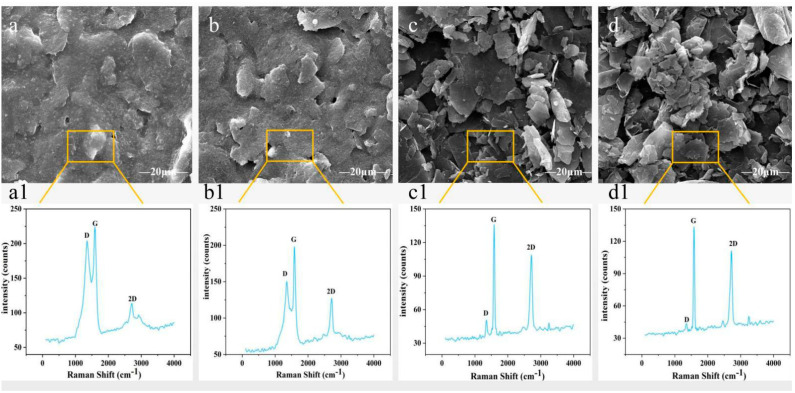
(**a**–**d**) SEM images of CCP/LIG with different LIG contents and (**a1**–**d1**) corresponding Raman spectra: (**a**) CCP + 0%LIG; (**b**) CCP + 3%LIG; (**c**) CCP + 5%LIG; and (**d**) CCP + 7%LIG.

**Figure 5 sensors-21-07022-f005:**
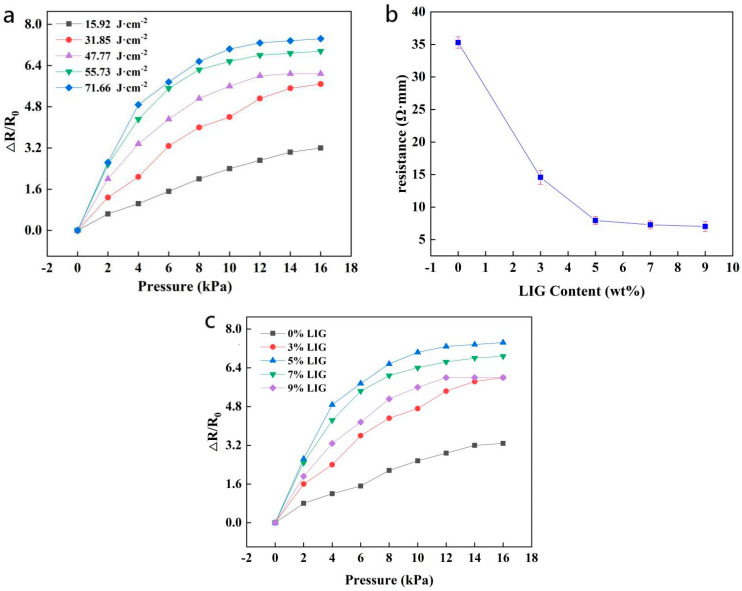
(**a**) The influence of templates processed with different laser fluxes on the sensitivity of the transferred CCP/LIG sensor; (**b**) Relationship between the LIG content and CCP/LIG resistivity; (**c**) The influence of LIG content in CCP/LIG on sensor sensitivity.

**Figure 6 sensors-21-07022-f006:**
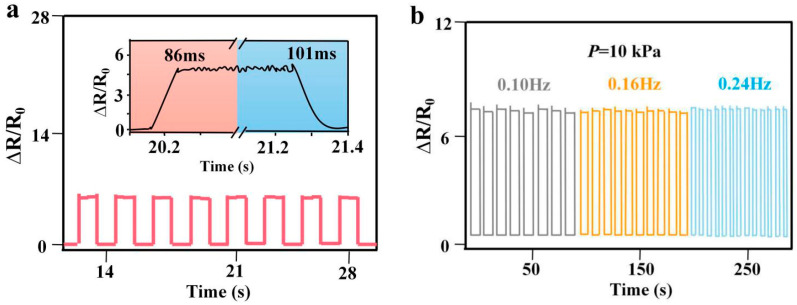
(**a**) Response time and recovery time of the CCP/LIG sensor; (**b**) relative resistance change of CCP/LIG at different frequencies at a pressure of 10 kPa; (**c**) CCP/LIG durability test at a pressure of 10 kPa.

**Table 1 sensors-21-07022-t001:** The relationship between sensor properties and machining parameters.

Group Machining Parameters	Mean Roughness (μm)	Bump Height (μm)	Scan Rate (mm/s)	Processing Frequency	Laser Injection Rate (J·cm^−2^)
Group A	19.8	140	500	10	15.92
Group B	36.8	200	500	10	31.85
Group C	53.6	230	500	10	47.77
Group D	59.4	260	500	10	55.73
Group E	77.7	330	500	10	71.66

## Data Availability

Not applicable.

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
