# Peer review of "Fabrication and Performance of Graphene Flexible Pressure Sensor with Micro/Nano Structure"

_sensors, 2021, doi:10.3390/s21217022_

Round 1

Reviewer 1 Report

Authors present big experimental and research work, what should be published but the way there are presenting their achievements requires corrections.  I will list my opinion in listed below points, some of them are just very minor remarks and only a small correction is required, but these kind of mistakes are irritiating and this why I have decided to list them. Only remark number 11 is important and for me crucial.

  1. Page 3 line 112 - what kind of PI was applied ?
  2. Page 3 line 121 - there are listed experiments with 3%, 5%, 7% and 9% of LIG added while on Fig. 4a; 4a1; Fig. 5c and later on page 8 line 259; 268; line 273 there are discussed results with 0% LIG added. And this is correct, that these experiments with 0% LIG were done.
  3. Page 3 line 128 - why as an elastic substrate Ecoflex was selected ? What is advantage and idea of using biodegradable film ? In sensor there is required reliability and durability and eg PET substrate should be used.
  4. Page 5 line 171 - should be sp2
  5. Page 5 line 179 - should be Fig.
  6. Page 3 line 103 Fig. 1 caption - should be a) b) c) as it is discussed below at line 112
  7. Page 6 line 210 - should be Jcm-2 in single line
  8. Page 8 line 247 - space is missing in "....resistivity (average value)....
  9. Page 10 line 342 - why only 650 cycles were done ? It was lasting too long ? No deterioration was observed, may be it will good to run for a longer period of time ?
  10. Page 10 line 316, 317 - remove this sentence
  11. Thare are listed properties of obtained pressure sensors. There are also listed other flexible pressure sensors (page 1 line 31). Why there is no comparison to other products, even without giving the producer   name ? Reader doesn't know after reading this paper - is it fantastic ? Or is it just poor ? Or maybe big experimental work was done and it is worth to publish it ? There is not such a comparison given.

Author Response

Dear reviewer,

Please see the attachment, thanks.

Reviewer 2 Report

The authors fabricated pressure sensors utilizing hybrid conductive carbon paste (CCP) and LASER-induced graphene (LIG) and evaluated their performance. I believe this study can be applicable for publication in Sensors, but several points listed below should be addressed:

  • [Title] Abbreviations (i.e. CCP and LIG) should not be used in the title. In addition, the title should contain “pressure sensor”.
  • [Abstract] Abstract seems too detailed and not well summarized. The authors should make it more concise so that the significance of this study is highlighted.
  • [l.66-] The logic flow of this paragraph is not clear. Why does “the materials and structure still need to be further improved”? Why is the micron-scale structure like taro leaves preferable for resolving the issue? (Are there any ways for it other than the micron-scale structure?)
  • [l. 105] The SEM image of a taro leaf should be included in the manuscript, and compare it with Figure 3.
  • [l. 210] Group A, B, C, D, and E are not defined while it seems the group corresponds to the LASER injection amount as shown in Figure 3. The explanation for the group should be clearly described in this paragraph.
  • [Figure 5b] What does the error bar represent? (standard deviation or standard error?) The authors should also describe the number of samples to evaluate the error bars.
  • [Figure 5] The hysteresis effect is an important factor in pressure sensors. Did the authors evaluate the effect? Either showing the data or commenting on it is necessary as long as this study is about pressure sensors.
  • [Conclusion] As the authors draw a perspective of using the pressure sensor in wearable medical monitoring equipment, intelligent robot technology, and human-machine interfaces, the authors also need to show the roadmap for that future; showing the requirements including sensitivity, response and recovery time, durability, for such applications, and compare them with the current performance of the CCP/LIG-based flexible pressure sensor. Also, comments on what should be further investigated or developed for the future make the paragraph more convincing.

Author Response

(The authors gave the same response as above.)

Reviewer 3 Report

The manuscript is well structured and easy to follow. The method hasbeen  described well. However, there are many works have been published in this field. I believe the novelty needs to be elaborated better in this manuscript.

More lit review (especially from recent published work for both sensor and fabrication method - especially the laser method) need to be added.

The statistical analysis need to be included as  well as number of samples and error bars in all graphs/tables.

Again, in conclusion, authors need to explain about the novelty of their results and the possible contributions of their work in this field.

Author Response

(The authors gave the same response as above.)

Reviewer 4 Report

Dear Authors,

I think that the manuscript (MS) must be improved and in some points clarified. I gave some suggestions in the following.

As a general comment, I recommend the Authors to reconsider all the content of the MS making it more clear and logic. They also must better describe the real novelty of this work.

Please also correct the English wording because in some points words are lacking and in others words are too many (taro leaf structure is repeated too much).

Please uniform the term Figures and their caption as in the Sensors template.

Pay attention to the term explore into the MS. Please substitute it (for example line 186).

Title

The title is very bad: sensors of what? ; what means preparation and performance research?.

Furthermore it contains 2 abbreviations.

Abstract

Abstract exceeds the 200 words. Please synthesize and make it more clear.

Line 20: aspect ratio here?

Introduction

Line 30: Smith et al.

Lines 41-44: please rewrite more clearly.

Line 53: Added without capital letter.

Line 70: the sentence cannot start with and. Moreover you must emphasize the concept that taro lives surface have many tiny needles which has a positive effect on improving the contact response of the flexible sensor. It a fundamental point of your research.

Line 77-78: I think that you must add at least a SEM image from which you prepare the template (here or better in the materials and methods section Lines 105-106).

Materials and Methods

Line 88: materials? methods? Or both?

Lines 89-90: please rewrite.

Lines 100-101: strain sensor? What is the advantage to cite figs 1 and 2 here. In my opinion you must better organize the content of this section (2.2) including the contents of the figures.

Lines 105-106: you must add at least a SEM image from which you prepare the template before Fig 1.

Line 106: here you have an aluminum alloy plate and not a template! Please pay attention to this point through the whole MS.

Lines 112-113: the images need a suitable description. The images’ numbers are inverted in this sentence.

Line 118: in this part, what part? Probably after…

Lines 124-125-126: I am sorry but until now you didn’t describe templates with different microstructures!

Line 128: Please add more details about Ecoflex.

Line 130: in fig 2b it seems that you use 2 CCP/LIG films for each sensor. If it is so you must add this information

Results

Lines 136-140: please rewrite more clearly.

Lines 142-145: it is not clear how many samples you have prepared.

Lines 145-146: in fig 3a there is only the case of laser flux of 71.66 J*cm-2.

Line 169: the results? Please better argument.

Lines 175-183: “In particular, 175 the D peak of low LIG content was particularly obvious, indicating more defects in the prepared samples, while the D peak of high LIG content (7%) was extremely weak, accompanied by better quality graphene. On the other hand, the sample with high content of LIG has an obvious 2D characteristic peak near the wave number of 2700 cm-1. In FIG. 4, the 2D peak of CCP/LIG with low content of LIG is less obvious than that of CCP/LIG with high content of LIG. Therefore, the 2D peak can be speculated to gradually disappear because of the excessive number of layers. A comprehensive judgment can be made that the CCP/LIG interlayer materials with high LIG contents are low layer graphene material with high quality.”

  • It is not obvious as the graphene quality can change by changing the graphene content. Your graphene is prepared all in the same way and nominally it has the same quality…
  • Please describe the 2D peak

Lines 186-189: please rewrite.

Lines 189-191: this sentence is no sense.

Lines 191-194: please rewrite. What is the meaning to write “sensor with a taro leaf structure” too many times? Sensor or pressure sensor is enough.

In the section 3.2.1 you start to divide the sensors in different GROUPS! You must add a table to clarify the kind of sensors in each group and their properties.

Line 216: please rewrite “all adjacent experimental group”.

Line 260: flat SEM morphology – you study the morphology with SEM microscope but the morphology is a property of the material and it is independent from the method that you use (SEM microscopy). So you can observe a flat morphology in the SEM image as reported in fig. 4. Please correct in all points.

Line 290: please add the method used to calculate the response and recovery times.

Line 305: in sum?

Conclusions

Lines 316-317: Please erase this sentence.

The conclusions are a summary of the results without any comments. Please rewrite the section.

Lines 346-349: Please add the motivations for all these very interesting applications.

Best regards

Author Response

(The authors gave the same response as above.)

Round 2

Reviewer 1 Report

This paper is well written and in my opinion is worth to be published. There is small correction required on page 7 lines 229; 230; 231.

Reviewer 2 Report

The authors well addressed the issues I suggested, but only one still remains unresolved. As the hysteresis effect is an essential matter in pressure sensors, the authors should comment on the effect. It does not matter if the data is shown in the manuscript or not. As many of the readers want to know the hysteresis effect of the sensors, just a simple comment like “further investigation on the hysteresis effect should be needed” makes sense.

Reviewer 4 Report

Dear Authors,

I thank you for your efforts to improve the manuscript (MS) following my suggestions.

Here, some more comments.

Line 30: “Smith et al.” the point is necessary in al. because it is an abbreviation.  

Fig 3: in my opinion the figure is not well presented. You must reorder the figures inside as C-A, B-B1 and better if you put them not in a raw but in a square where C-A are above B-B1.

Line 172: Better “SEM image of a taro leaf surface.”

Line 169: the results? Please better argument.

Lines 175-183: “In particular, 175 the D peak of low LIG content was particularly obvious, indicating more defects in the prepared samples, while the D peak of high LIG content (7%) was extremely weak, accompanied by better quality graphene. On the other hand, the sample with high content of LIG has an obvious 2D characteristic peak near the wave number of 2700 cm-1. In FIG. 4, the 2D peak of CCP/LIG with low content of LIG is less obvious than that of CCP/LIG with high content of LIG. Therefore, the 2D peak can be speculated to gradually disappear because of the excessive number of layers. A comprehensive judgment can be made that the CCP/LIG interlayer materials with high LIG contents are low layer graphene material with high quality.”

  • It is not obvious as the graphene quality can change by changing the graphene content. Your graphene is prepared all in the same way and nominally it has the same quality…
  • Please describe the 2D peak

Reply: Thanks very much for your careful check and valuable advice. According to your advice, we quite agree with your statement that "It is not obvious as the graphene quality can change by changing the graphene content". We re-conducted Raman spectrum analysis of LIG/CCP film, including other samples with different laser injection amounts, and obtained the same conclusion as above. We are very grateful for your comments and have revised this part of the description. Further, we will pay close attention to this problem in future experiments. Thanks.

My comment was not about the assessment of the credibility of your results. I disagree with the use of term obvious for the results as well as to the lacking in descriptions. You must substitute the term obvious with the motivation of the results.

Line 228: in the table, processing times lack of measuring unit.

Best regards
